# THE POWER OF TASK-BASED APPROACH IN BUILDING TRUSTWORTHY AI SYSTEMS

**Evgenii Vityaev**[1,2], **Sergey Goncharov**[2], **Dmitry Sviridenko**[1] **and Andrey Nechesov**[1] *

[1] Artificial Intelligence Research Center of Novosibirsk State University, Novosibirsk, Russia
[2] Sobolev institute of mathematics SB RAS

`vityaev@math.nsc.ru, s.s.goncharov@math.nsc.ru;`
`dsviridenko47@gmail.com; nechesoff@gmail.com;`

## ABSTRACT

Artificial intelligence systems are now integral to virtually every facet of our lives, exhibiting an ability to reason and solve problems within defined formal frameworks. However, challenges remain, particularly the issue of hallucination—where AI systems generate incorrect or misleading information. This paper proposes a task-based approach to building reliable AI systems, focusing on the task itself and the criteria necessary for its resolution. Our objective is to ensure that AI systems not only provide solutions but also possess an understanding of the underlying limitations of the problem. This includes identifying the axioms and theorems involved, allowing the solution process to be informed by a clear comprehension of the problem's structure and constraints.

## 1 INTRODUCTION

Today, modern large language models have advanced far in their cognitive capabilities. This allows you not only to ask questions and get answers to them, but also to analyze the final result itself directly. By setting the logic for LLM systems, we can also change the final result. For example, if you ask ChatGPT to calculate an infinite sum of numbers like:

$$1 + 2 + 4 + 8 + ... + \tag{1}$$

then it will first suggest using the sum of geometric progressions of the first n terms of this sequence, and the second option for an infinite sum will give the answer infinity. But if the same ChatGPT is asked to calculate the same infinite sum (1) using the Ramanujan method, it will give the answer -1. This is achieved due to the fact that if we denote the final sum for S, it is easy to see that 2S-S = -1. From this we get that S = -1. Although this value does not make sense in classical mathematics, it does make sense as an analytic extension of a complex function.

By changing requests as LLM tasks, we also get a different response, setting new task conditions. To understand when LLM uses certain reference rules, we need to develop a clear mechanism for solving the tasks set.

This is exactly what the task-based approach can help with. This approach was proposed back in the 30s of the last century, but was rather fully formalized by academician Yu. L. Ershov and professor K. F. Samokhvalov in their book «New Philosophy of Mathematics, illnesses and treatment» Ershov & Samokhvalov (2007). An important point is that the task is defined only when the criterion for its solution is set. If we do not have a criterion for solving the task, then there is no task, because any text can be considered as a solution.

*This work was supported by a grant for research centers, provided by the Analytical Center for the Government of the Russian Federation in accordance with the subsidy agreement (agreement identifier 000000D730324P540002) and an agreement with the Novosibirsk State University dated 27 December 2023 No. 70-2023-001318.
Evgenii Vityaev and Sergey Goncharov thanks for financial support the State Assignment "Logical calculus and Semantics, Model theory and Computability" FWNF-2022-0011

Moreover, most often it is also important to us that when solving tasks, we are given not only the answer that meets the criterion of success, but also how it was obtained. In other words, the process of obtaining the result itself is important. We are very pleased that in recent months, large language models have learned to explain the solution process itself. It is clear that these are only the first results and it is necessary to conduct a thorough analysis of their reasoning, but the very fact that this is possible shows great prospects for their use and increasing confidence in them.

This will allow them to be used in areas where some governments have restricted the use of LLM at the moment: finance, medicine, scoring, etc. In general, the topic of trusted artificial intelligence is more relevant than ever. And the key to all this is the topic of security of their use in various areas.

Another important area of development within the framework of trusted AI is the development of hybrid artificial intelligence, a combination of LLM, logical-probabilistic approach and other cognitive methods, which allows combining the advantages of large language models with the correctness of logical reasoning. Such techniques on logical-probabilistic inference and learning theory are presented in the works of the authors of this paper.

## 2 TASK-BASED APPROACH

As will be argued later, the basis of purposeful activity is the task solution. Therefore, it is important to be able to correctly formulate tasks and set criteria for their solution. If we refer to the work of Ershov and Samokhvalov, they define the task as a certain logical formula and the solution of the task is the correct interpretation of the free variables under which this formula becomes true Ershov & Samokhvalov (1984). In a more general case, the task solution is a computable term for which the values of free variables in the formula are searched using fixed interpretation of free variables of the term, at which the logical formula becomes true. In general, the task concept also was considered back in the 30s by Kolmogorov for the intuitionistic calculus of statements.

Thus, the works of Kolmogorov, Ershov and Samokhvalov served as an impetus for the formalization of the task-based approach, in which the key role is played the task and the criterion of its solution. It is assumed that AI systems should be designed to solve specific tasks with an emphasis on their explicability and purposefulness. This approach combines elements of the agent-based approach successfully applied in AI and the principles of general artificial intelligence (AGI), allowing you to model complex decision-making processes.

This paper provides a rationale for the task-based approach as a conceptual framework for creating trustworthy artificial intelligence. Key techniques for implementing the principles of the task-based approach in modern intelligent systems are considered. The main focus is on how this concept can contribute to increasing user confidence in AI and adapting technologies to meet real human cognitive needs.

Given such a huge interest in the problem of trust and explainability of AI, the authors of this paper present to the public their original concept of **building trustworthy artificial intelligence (AI)**, which is based on the task-based approach to artificial intelligence Ershov & Samokhvalov (1984; 2007); Goncharov et al. (2020a;b); Sviridenko (2019); Sviridenko & Vityaev (2020); Vityaev et al. (2023), which generalizes to a certain extent such well-known and popular approaches as the agent approach and general artificial intelligence (AGI).

Note that the success and popularity of the **agent-based approach**, described in the monograph Russell & Norvig (2006) are largely due to the well-chosen system of basic concepts, which includes various types of "agents" and "external environment". Various tasks of artificial intelligence are considered in this approach as tasks of interaction between "agent" and "environment" which allowed its authors to carry out a detailed classification of agents and environments. From our point of view, this methodology lacks an even more important and general concept that covers both agents and environments – this is the concept of the task that agents solve in their respective environments. Thus, the agent approach actually follows the task-based approach, but only without explicitly defining the tasks solved by agents and specifying all the components inherent in the concept of "task".

**General Artificial Intelligence** (Artificial General Intelligence, AGI), reviewed in Vedyakhin (2021), has shown that modeling human cognitive processes is not a prerequisite for solving intellectual problems. Therefore, AGI reflects the fact that artificial intelligence can be possessed to

one degree or another by both a person or a living organism with a highly developed central nervous system, and an abstract robotic system. Leading AGI developers (Ben Hertzel, Shane Legge, Pei Wang) define AGI as follows: "it is the ability to solve cognitive tasks in general, acting purposefully, adapting to environmental conditions through training, minimizing risks and optimizing losses to achieve their goals" Vedyakhin (2021). There is currently no unified methodology for developing AGI.

## 2.1 Formalization of tasks using the task-based approach

We have developed a task-based approach to AI, which covers both the tasks solved by agents and the AGI task formulated above. At the same time, the task-based approach is trustworthy and explicable, since it has the following important properties:

1. The task to be solved is set within the Subject Domain (SD) ontology, along with the data and knowledge used.

2. The task to be solved is formulated in the SD ontology as a request to the SD model.

3. The request is formulated in terms of the specifications that can be executed on the SD model. Task specifications generate algorithms for solving them.

4. The received response to a request that provides a solution of the task is checked by a special criterion, formulated together with the task in the SD ontology, which checks that this answer is indeed a task solution.

5. For some types of specifications, the polynomial computability of the task-solving algorithms generated by them is proved.

6. Specifications may include reference to oracles represented by certain predicates or functions that can be calculated by different algorithms, including neural networks.

7. The process of task solving is anthropomorphic, because it follows the cognitive process of purposeful behavior in accordance with the Theory of Functional Systems of the brain activity [15-20].

## 2.2 Cognitive modeling in a task-based approach.

The task-based approach covers the AGI goal formulated above, as it provides a formal model of cognitive purposeful activity based on the Theory of Functional Systems of brain activity Anokhin (1978; 1974); Vityaev (2015; 2021)

A generalization of the task concept in cognitive sciences is the concept of Goal Anokhin (1978; 1974); Vityaev (2015). A goal cannot be achieved without a criterion for its achieving, otherwise it can always be assumed that it has already been achieved. Therefore, the Goal statement should always include a criterion for achieving the goal, just like for the task.

Currently, the only physiological theory that considers Goal achievement as the brain's solution of the task of satisfying a certain need is the Theory of Functional Systems Anokhin (1978; 1974; 1984); Vityaev (2014; 2015; 2021). This theory also reveals the physiological mechanisms of goal achievement and task solving by the brain: "Perhaps one of the most dramatic moments in the history of the study of the brain as an integrative education is the fixation of attention on the action itself, and not on its results ... we can assume that the result of the "grasping reflex" will not be the grasping itself as an action, but that set of afferent stimuli that corresponds to the signs of the "grasped" object" Anokhin (1984). The "set of afferent stimuli" is the criterion for achieving the goal in the TFS.

In Vityaev (2014; 2015; 2021); Mukhortov et al. (2012), a formal model of TFS was developed, which is an integral part of the task-based approach. This model was successfully used for modeling animates Vityaev (2015); Mukhortov et al. (2012); Demin & Vityaev (2014); Vityaev et al. (2020).

# 3 MATHEMATICAL THEORY OF THE TASK-BASED APPROACH - SEMANTIC MODELING

In order to correctly formalize tasks and correctly solve them, it is important to choose the correct syntactic constructions and correct formalization for the subject area. First of all, we want to solve tasks quickly. For solving tasks, an acceptable characteristic is the polynomial computational complexity in time. If a certain task is solved in an exponential time from the length of the input data or is not solved at all, then such approaches are most often not interesting or feasible in practical terms.

Semantic programming is best suited for this purpose. The subject area is represented as a hereditary-finite list superstructure of the form $HW(\mathfrak{M})$ of some finite signature $\sigma$. This model has proven itself very well in practice. First of all, this concerns the property that if $\mathfrak{M}$ of the signature $\sigma_0$ is polynomial-computable, then $HW(\mathfrak{M})$ of the signature $\sigma = \sigma_0 \cup \{\in^{(2)}, \subseteq^{(2)}, U^{(1)}, nil\}$ will be also polynomial-computable. Due to the fact that the add-in contains lists and defines relationships to be an element of the list or its initial segment. This allows us to define a set of Delta-0 formulas, which is given inductively and in which all quantifiers of existence and universality are bounded. That is, we get a limited search over the elements of lists or their initial segments, which allows us to guarantee that checking the truth of such formulas on HW(M) will have polynomial computational complexity. Moreover, new termal constructions can be constructed: conditional terms, p-iterative terms, and so on, which guarantee that the termal expansion of our set of formulas will be conservative.

First of all, we need to highlight a number of important results that will help us correctly formalize problems and find solutions to them:

1. Polynomial Analogue of Gandy's fixed point theorem (PAG-theorem) Goncharov & Nechesov (2021). In this paper, we construct a special operator whose smallest fixed point is polynomial-time computable (p-computable).

$$\Gamma^{HW(\mathfrak{M})}_{P_1^+,\ldots,P_n^+}(\Gamma^*) = \Gamma^* \tag{2}$$

where $\Gamma^* = (Q_1^*, \ldots, Q_n^*)$ is a smallest fixed point where $Q_i^*$ - the set of the truth for predicate $P_i$.
This allows us to define inductively definable constructions, the set of which will be the smallest fixed point which will be a p-computable.

2. Solving the P=L problem Goncharov & Nechesov (2022a). This result allowed us for the first time to construct a p-complete logical programming language in which the program has a special term. This result guarantees us that the language's expressive power is sufficient to implement any algorithm of polynomial complexity.
Mathematically it can be explained as follows: let $f$ some p-computable functions, then there exists a suitable Turing machine $M$ implementing $f$. The machine $M$ has a fixed program $P_M$, according to this program $P_M$ we form a suitable p-iteration term $t$ which calculates exactly the same thing as the p-computable function $f$.

3. Functional variant of the polynomial analogue of Gandy's fixed point theorem (FPAG-theorem) Goncharov & Nechesov (2024). The same result as for PAG-theorem, but now we guarantee that any recursive function constructed using the operator from the conditions of the FPAG-theorem will have polynomial computational complexity. Further, you can use these functions to enrich our p-complete language $L$ and this extension will be conservative. Now this operator, unlike (equation 2), acts on the space of functions:

$$\Gamma^{HW(\mathfrak{M})}_{f_1^+,\ldots,f_n^+}(F^*) = F^* \tag{3}$$

where $F^* = (f_1^*, \ldots, f_n^*)$ is a smallest fixed point and $f_i^*$ this is a p-computable continuation of the function $f_i$ respectively.

4. Methodology of programming in Turing-complete languages Goncharov et al. (2024). Using the first 3 results, we can now isolate a polynomial fragment of the Turing-complete language, which guarantees us polynomial computational complexity. This creates a programming methodology that can be used in any programming language that meets the initial conditions.

5. It should be noted that if a certain system is p-computable Goncharov & Nechesov (2023b); Nechesov (2023b), then there exists a polynomial-computable representation for it in a suitable p-computable hereditary-finite list superstructure HW(M) on the p-complete language $L$.

6. Another important tool is the Learning Theory and Knowledge Hierarchy for Artificial Intelligence Systems Nechesov (2023a). Where the concept of probabilistic knowledge is introduced and a hierarchy of probabilistic knowledge is defined. This allows us to instantly select the most effective probabilistic knowledge from the database for further use. This approach guarantees us confidence in the correctness of logical reasoning based on the probabilistic knowledge that is available in the knowledge base.

7. Of course, it is worth considering the work on the combination of AI and blockchain technologies. The axiomatization of blockchain Goncharov & Nechesov (2023a) allowed working with these structures at a logical level, calculating complexity, building multi-blockchains. It was the unification of the two technologies that set the direction for the implementation of the framework for civil participation in the management of smart cities Nechesov & Ruponen (2024).

8. The task approach helped in the formation of collective intelligence for multi-agent systems in virtual cities Nechesov et al. (2025), in which there are two types of agents based on LLM and logical-probabilistic agents that control the work of the former. This hybridization places great hopes on the task-based approach in the construction of MAS systems and their development in virtual cities.

## 4   INDUCTIVE INFERENCE OF KNOWLEDGE IN THE TASK-BASED APPROACH

The Subject Domain (SD) can be defined as empirical system $\Im = \langle A, \Omega \rangle$, where A – is the objects of the subject domain, and $\Omega$ – is the *domain ontology* of SD – the set of all relations and operations interpreted in SD Kovalerchuk & Vityaev (2000); Vityaev & Kovalerchuk (2008; 2017). It is necessary for the trust approach to AI that a person understand and interpret the ontology of the subject domain.

Inductive inference of knowledge – is a generalization of individual cases into general statements that may be applied to other cases. Inductive inference of knowledge by some machine learning method, must be able correctly process the objects properties and attributes in order to obtain knowledge that interpretable in the ontology of SD.

Let us consider the problem of the empirical system theory $Th(\Im)$ discovery. We assume that the theory of $Th(\Im)$ is a collection of the universal formulas (a more general case considered in Vityaev (2017)). It is known that a set of universal formulas is logically equivalent to the set of rules of the form:

$$\forall x_1, \ldots, x_2 (A_1 \& \ldots \& A_k \Rightarrow A_0), \ k \ \geq \ 0, \tag{4}$$

where $A_0, A_1, ..., A_k$ are literals. Therefore, we can assume that the theory $Th(\Im)$ is a set of rules (4).

It is easy to see that the rule $C = (A_1 \& \ldots \& A_k \Rightarrow A_0)$ logically follows from any of its *sub-rules* of the form: $(A_{i1} \& \ldots \& A_{in} \Rightarrow A_0)$, where $\{A_{i1}, ..., A_{in}\} \subset \{A_1, ..., A_k\}$, $0 \leq n < k$ and $(A_{i1} \& \ldots \& A_{in} \Rightarrow A_0) \vdash (A_1 \& \ldots \& A_k \Rightarrow A_0)$. Then the theory of $Th(\Im)$ can be simplified. The *law* of the empirical system $\Im = \langle A, \Omega \rangle$ is a rule C of the form (4) that is true on $\Im$ but every its sub-rules is not true on $\Im$. Let $L$ be the set of all laws. Then it can be proved that $L \vdash Th(\Im)$ Vityaev (2014; 2006). In this case, the theory $Th(\Im)$ can be considered as a set of laws of an empirical system.

Let us define the probability $\eta$ on empirical system $\Im = \langle A, \Omega \rangle$ as on the model Halpern (1990). By the *probabilistic law* on $\Im$ we define the rule $C = (A_1 \& \ldots \& A_k \Rightarrow A_0)$ for which the conditional probability $\eta(A_0 \& A_1 \& \ldots \& A_k) / \eta(A_1 \& \ldots \& A_k)$ is defined $(\eta(A_1 \& \ldots \& A_k) > 0)$ and strictly more than the conditional probabilities of each of its sub-rules. By the *Strongest Probability Law* (SPL) we mean the probabilistic law C, which is not a sub-rule of any other probabilistic law.

Inductive inference of probabilistic domain knowledge can be sufficiently fully realized by the following semantic probabilistic inference.

We will call the sequence $C_1 \sqsubset C_2 \sqsubset ... \sqsubset C_n$, $C_i = (A_1^i \& ... \& A_{k_i}^i \Rightarrow G)$ of probabilistic laws a *Semantic Probabilistic Inference* (SPI) of some strongest probabilistic law $C_n$ predicting some fact $G$ and starting from $C_1 = (\Rightarrow G)$ and every rule $C_i$ is a sub-rule of the rule $C_{i+1}$ and $\eta(C_i) < \eta(C_{i+1})$, i = 1,2, ... n-1.

Knowledge is need for predictions. We now define the strongest probabilistic laws that solve the problem of statistical ambiguity Vityaev (2006); Vityaev & Odintsov (2019) and predict without contradictions.

Let us consider the set of all strongest probabilistic laws predicting some fact G together with their semantic probabilistic inferences. This set can be represented by a *semantic probabilistic inference tree* of the fact G.

We will call the strongest probabilistic law belonging to the semantic probabilistic inference tree of the fact G, which has the maximum value of the conditional probability among all the rules of the tree, the *Most Specific Probabilistic Law* of inference G (MSPL(G)). The set of all maximally specific laws MSPL(G) for all literals G $\in \Omega$ we denoted as MSPL.

It can be proved that L $\subseteq$ MSPL Vityaev (2006; 2017) and therefore the set of laws MSPL generalize the theory $Th(\Im)$ and includes not only rules that are true on $\Im$, but also probabilistic ones. At the same time MSPL, like any theory, is logically consistent Vityaev (2006; 2017); Vityaev & Odintsov (2019) and therefore, in the exact sense a ***probabilistic theory*** of the subject domain $\Im = \langle A, \Omega \rangle$.

This method of the inductive knowledge discovery on the empirical system $\Im$ is implemented in the form of the platform and software system "Discovery", described below. It was successfully applied to solution of many practical tasks (see Scientific Discovery website http://old.math.nsc.ru/AP/ScientificDiscovery/index.html).

## 5 PREDICTIONS IN THE TASK-BASED APPROACH

We will prove that the predictions based on MSPL laws are consistent. In the philosophy of science predictions are described by the so-called Covering Law Models (Britannica), which consist in deducing facts as special cases of laws. There are two prediction models:

1. Deductive-Nomological (D-N), based on facts and deductive laws.
2. Inductive-Statistical (I-S), based on facts and probabilistic laws.

The deductive-nomological model can be represented by the following scheme:

$$\frac{L_1, ..., L_m}{\dfrac{C_1, ..., C_n}{G}}$$

where:

1. $L_1, ..., L_m$ – set of laws;
2. $C_1, ..., C_n$ - set of facts;
3. $G$ – predicted statement;
4. $L_1, ..., L_m, C_1, ..., C_n \vdash G$;
5. Set $L_1, ..., L_m, C_1, ..., C_n$ is consistent;
6. Laws $L_1, ..., L_m$ contain only generality quantifiers;
7. Facts $C_1, ..., C_n$ – quantifier-free formulas.

**The inductive-statistical model** is similar to the previous one, with the difference that property 6 is formulated differently and the *Requirement of Maximum Specificity* (RMS) is added:

6. The set $L_1, ..., L_m$ contains statistical laws.

RMS: All laws $L_1, ..., L_m$ are maximally specific.

According to Hempel Hempel (1965; 1968) (RMS) is defined as follows. The following I-S inference

| p(G;F) = r | |
|---|---|
| F(**a**) | [r] |
| G(**a**) | |

is acceptable in the state of knowledge K if for each class H, for which both of the following two statements belong to K: $H(x) \subset F(x)$, $H(\boldsymbol{a})$, there is a statistical law $p(G; H) = r'$ in K such that $r = r'$.

The RMS requirement is that if F and H both contain object **a**, and H is subset of F, then H has more specific information about the object **a** than F, and therefore the law p(G;H) should be preferred to the law p(G;F). However, the law p (G;H) must have the same probability as the law p (G;F).

***The problem of statistical ambiguity and its solution***. In the process of inductive statistical inference, we can obtain statements from which contradictory may be derived. Hempel hoped to solve this problem by requiring statistical laws to satisfy the requirement of maximum specificity, but he and his followers did not prove that there would be no contradictory conclusions in this case.

We present a maximum specificity requirement that generalizes the RMS, for which we prove the inconsistency of inductive statistical inference. We assume that the class H of objects in the RMS definition is defined by some statement H in the ontology $\Omega$.

**Requirement of maximal specificity** Vityaev (2006): If you add any statement H to the premise of the rule $C = (F \Rightarrow G)$ (note that in this case the statement $\forall x(F(x)\&H(x) \Rightarrow F(x))$ is true) and $F(a)\&H(a)$ is true, then the equality $h(G/F\&H) = h(G/F) = r$ must be fulfilled.

**Theorem** Vityaev (2006). Any law from MSPL meets the RMS requirement.

Therefore, any maximally specific probability law satisfies the requirement of maximum specificity RMS.

If the most specific rules from Hempel's definition are understood as the most specific probabilistic laws, then the problem of statistical ambiguity is solved by virtue of the following theorem.

**Theorem** Vityaev (2006); Vityaev & Odintsov (2019). The I-S inference is consistent if applying to any theory $T \subseteq MSPL$.

## 6   INTELLIGENT LEVEL OF AI SYSTEMS: COMPARISON

Using the obtained results, we can determine the Intelligent level for AI systems in relation to a given theory $T$ with fixed model of the theory $\mathfrak{M}$ and a base of logical-probabilistic knowledge $K$ of this theory $T$.

Let $A$, $B$ be some intelligent systems then we can compare intelligence levels $IL(A)$ and $IL(B)$ relative to the set $S$ within the framework of a theory $T$, its model $\mathfrak{M}$ and probabilistic knowledge base $K$.

Let $S$ be a set of tasks of the theory $T$ for which there is some solution within the framework of the theory $T$ and probabilistic knowledge from the base $K$.

Consider all tasks from the set $S$ have the following form:

$$\varphi : \ \forall x \exists y \Phi(x, y) \to \Psi(x, y) \tag{5}$$

where the formulas $\Phi$ and $\Psi$ have the form of a conjunction of literals $A_i$.

We will also assume by default that we have some simplifications of the $\phi$ in $S$ of the form:

$$\varphi : \ \forall x \in t \exists y \Phi(x, y) \to \Psi(x, y)$$

or

$$\varphi: \ \forall x \in t_1 \exists y \in t_2(x) \Phi(x, y) \rightarrow \Psi(x, y)$$

or

$$\exists y \Phi(\overline{c}, y) \rightarrow \Psi(\overline{c}, y)$$

Probabilistic solution for equation 5 will be a term $y = t(x)$ that makes the formula $\varphi$ true with some probability p ($\models^p$):

$$\mathfrak{M} \models^p \varphi(x, t(x)) \tag{6}$$

We will say that one system $A$ solved the task $s \in S$ better than another $B$ relative $K$ if the probability $p_A$ is better than the probability $p_B$ for their probabilistic solutions $t_A$ and $t_B$, respectively with hints from the knowledge base $K$. If some intelligent system has found a solution better than the strongest solution in $K$, then it is recorded in the knowledge base of $K$.

It is possible to define a relationship $\leq_{S,K}^{\mathfrak{M}}$ of the form:

$$IL(A) \leq_{S,K}^{\mathfrak{M}} IL(B) \Leftrightarrow n(A|B)_{S,K}^{\mathfrak{M}} \leq n(B|A)_{S,K}^{\mathfrak{M}}$$

where $n(A|B)_S^{\mathfrak{M}}$ the number of problems that were solved $A$ better than $B$ within the framework of tasks from set $S$ on the model $\mathfrak{M}$ with probabilistic knowledge base $K$. We assume that the sequence of incoming tasks $s_1, \ldots, s_n$ from $S$ is the same. The systems operate autonomously.

Proposition: The relation $\leq_{S,K}^{\mathfrak{M}}$ is an order.

To prove this necessary to check the axioms defining the order (reflexive, transitive, antisymmetric).

Let us have the finite sets of tasks $S_1, \ldots, S_n$ from different fields of knowledge with unique theories, models and knowledge bases, then we will say that one intelligent system $A$ is totally better than another $B$, if:

$$\forall i \ IL(A) \leq_{S_i, K_i}^{\mathfrak{M}_i} IL(B)$$

This approach helps to formalize the comparison of the intellectual capabilities of various large language models and, moreover, make them consistent with the probabilistic knowledge base $K$, which allows use LLMs in various fields in the future within the framework of trustworthy artificial intelligence.

## 7    PLATFORM SOLUTIONS OF THE PROBLEM APPROACH

Currently, semantic modeling, as one of the concepts of automatic solution of intellectual tasks, is based not only on the methodology and theory of the task-based approach, but also has at its disposal a well-developed toolkit aimed at supporting and maintaining the following technological scheme for solving intellectual tasks.

- STEP 1. The need that has arisen is clarified, the contradiction associated with it is identified and studied, related to the lack of a "template" way to meet this need. Note that this contradiction, in its essence, is the true reason and content of the Task being solved. Next, a criterion for the success of overcoming the identified contradiction is formulated in natural language (a prototype of the criterion for the task solution), if necessary, a decomposition of the contradiction task is carried out and the context of the task is determined (why, how, the nearest subtasks, the purpose and consequences of solving/not solving the task, etc.).

- STEP 2. General knowledge relevant to the problem is described in natural language, an ontological model of the problem domain (concepts, facts, rules, relations, ...) is constructed, and a class of queries for the problem domain is formulated in general and ontological terms.

- STEP 3. In logical-probabilistic terms of semantic modeling, a formal model of the task domain is constructed, and in corresponding terms, the query and the solution criterion are formulated.

- STEP 4. A computer model of the task is built within the framework and by means of the corresponding instrumental platform of semantic modeling.

- STEP 5. A computer model is used to find answers to queries. The feasibility of the decision criterion, the validity and/or explainability of the answers and their interpretability are checked.

As for the technological tools of semantic modeling, several platform have been created and are actively developing.

- The **D0SL platform** was developed under the leadership of V.S.Gumirov and allows you to control the logic of the behavior of complex systems using the d0sl language, understandable to a specialist in the subject area Gumirov et al. (2018). The platform has a wide range of applications, from enterprise business processes to project management or the behavior of autonomous systems, including artificial intelligence systems and the Internet of Things.

- The **bSystem platform** Mantsivoda & Ponomaryov (2019; 2020); Kazakov et al. (2020) is a platform for building digital counterparts of organizations and processes. It has been developed by the A.V.Mantsivoda research group for a number of years. The platform is focused on creating intelligent management systems for large business ecosystems, digital transformation of enterprises and other integrated solutions.

- The **Discovery system** was developed under the guidance of Professor E.E.Vityaev and allows you to identify patterns and predict events Kovalerchuk & Vityaev (2000); Vityaev & Kovalerchuk (2008; 2017). The Discovery system discovers knowledge in terms of the subject domain ontology. Interpretability of the produced patterns is very important when making responsible decisions in areas such as medicine, finance, or military applications.

- The **Delta platform** is a platform for the implementation and execution of programs written using Delta's special p-complete language on a virtual Delta machine Goncharov & Nechesov (2022b); Dolgov (2023). This logical p-complete language was developed by a group of leading mathematicians of the Siberian school: academician S.S. Goncharov, Professor D.I.Sviridenko, Dr. Nechesov and Master of Mathematics Dolgov. The platform is scalable and also allows you to connect logical learning modules of intelligent systems.

## 8  CONCLUSION

This paper can be considered as a report on the results of research in the field of the task-based approach and its practical applications carried out by the team of authors of the paper, their students and colleagues, which were started back in the 70s and 80s of the last century. Over the years, a lot of work has been done to develop and practically apply the provisions of the task-based approach in relation not only to various problems of the foundations of mathematics and AI, but also to such areas as cognitive sciences, neurophysiology, medicine, digital transformation of enterprises, automation of design of complex systems, digital twins, etc. As can be seen from the above, it was possible not only to significantly develop the methodological and mathematical provisions of the problem approach, but also to create its effective instrumental and technological base, which made it possible to successfully solve of a wide range of tasks from various fields: telecommunications, business, retail, fintech, genetics, geology, cybersecurity, robotics, medicine, etc. The creation of a trust–based "General" AI is on the agenda. And the experience of previous years shows, there is every reason that such an AI will be created in the near future. Including within the framework and by means of the task-based approach.

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
