# OpenReview forum: "The Power of Task-Based Approach in Building Trustworthy AI Systems"
_mathai.club/MathAI/2025/Conference — MathAI 2025 Oral_

### Official Review · Reviewer_EXG9 · 2025-02-26
**THE POWER OF TASK-BASED APPROACH IN BUILDING TRUSTWORTHY AI SYSTEMS. Reviewer recommends to include it in the Program of the International conference “Mathematics of Artificial Intelligence” (24-28 March 2025, Sochi) with its publication.**

**Rating:** 9
**Confidence:** 4

**Review:**

Trust and explainability of AI is one of the most important problems. In particular, it concerns the issue of hallucination—where AI systems generate incorrect or misleading information.The authors of the article present their original concept of building trustworthy artificial intelligence (AI), which is based on the task-based approach to artificial intelligence in works by Ershov & Samokhvalov (1984; 2007); Goncharov et al. (2020a;b); Sviridenko (2019); Sviridenko & Vityaev (2020); Vityaev et al. (2023). This approach generalizes to a certain extent such well-known and popular approaches as the agent approach and general artificial intelligence (AGI).  This article proposes a task-based approach to building reliable AI systems, focusing on the task itself and the criteria necessary for its resolution. The authors’s objective is to ensure that AI systems not only provide solutions but also possess an understanding of the underlying limitations of the problem. This includes identifying the axioms and theorems involved, allowing the solution process to be informed by a clear comprehension of the problem’s structure and constraints. The authors have developed a task-based approach to AI, which covers both the tasks solved by agents and the AGI task formulated in the article. The task-based approach covers the AGI goal formulated above, as it provides a formal model of cognitive purposeful activity based on the well-known Theory of Functional Systems of brain activity.
    The article presents the results of research in the field of the task-based approach and its practical applications carried out by the team of its authors, their students and colleagues over the course of several decades to create an appropriate effective instrumental and technological base, which made it possible to successfully solve of a wide range of tasks from various fields: telecommunications, business, retail, fintech, genetics, geology, cybersecurity, robotics, medicine, etc. The creation of a trust–based ”General” AI is on the agenda with using the task-based approach.
     The wide list of references in the article includes more that 50 appropriate literary sources. The quality, clarity, originality and significance of this work are high.
      I recommend to include it in the Program of the International conference “Mathematics of Artificial Intelligence” (24-28 March 2025, Sochi) with its publication.

---

### Official Review · Reviewer_5F68 · 2025-02-27
**Thorough theoretical work**

**Rating:** 8
**Confidence:** 3

**Review:**

The article describes the application of the task-based approach for the realization of trusted artificial intelligence. In the sections of mathematical basis of the approach it is shown how the comparison of intellectual capabilities of different LLMs can be formalized. Accordingly the formalization of the stating the problem and the criterion of its solution in the issue of building trusted AI is shown.

For all the advantages of this work, there are still **several remarks:**
- Section 3 lists a number of important results, as far as can be understood, prefacing and supporting this work. It is not clear how paragraphs 7 and 8 on blockchain technologies, smart city and multi-agent systems are related to the overall theme of the paper.
- On page 5 there is a statement “Knowledge is needed for predictions” -- is this the author's opinion or a quote?
- The style of writing in the abstract and in the introduction has to be checked.

**Strengths of the article:**
Good theoretical background. Enumeration of technological tools of semantic modeling in section 7. If there are references to their implementation, you could add them.

---

### Official Review · Reviewer_TqN3 · 2025-02-27
**Intresting work for area how and what AI learns. Accept**

**Rating:** 8
**Confidence:** 3

**Review:**

The article describes the task-oriented approach as a key tool for ensuring the reliability of AI, justifies it with theoretical premises and illustrates it with mathematical theory. A promising approach is presented, where tasks determine how and what AI learns, which makes its work more transparent and predictable. This is especially important in areas such as medicine and finance, where the cost of error can be too high.

The main drawback of the article is the lack of specific examples of practical application of the methods proposed in it

---

### Decision · Program_Chairs · 2025-03-08

**Decision:**

Accept (Oral)

**Comment:**

Your article has been accepted and you can give a talk on the article. All articles will be sorted by rating and within the available conference places one author from each article will be invited. If there are not enough places, then you will either have the opportunity to speak remotely or come at your own expense!